# Characterization of Biocomposites and Glass Fiber Epoxy Composites Based on Acoustic Emission Signals, Deep Feature Extraction, and Machine Learning

**DOI:** 10.3390/s22186886

**Published:** 2022-09-13

**Authors:** Tomaž Kek, Primož Potočnik, Martin Misson, Zoran Bergant, Mario Sorgente, Edvard Govekar, Roman Šturm

**Affiliations:** 1Faculty of Mechanical Engineering, University of Ljubljana, 1000 Ljubljana, Slovenia; 2LTH Castings d.o.o., 4200 Škofja Loka, Slovenia; 3Optics11, 1101 BM Amsterdam, The Netherlands

**Keywords:** polymer composites, biocomposites, GFE composites, acoustic emission, deep feature extraction, convolutional autoencoder, machine learning, neural networks

## Abstract

This study presents the results of acoustic emission (AE) measurements and characterization in the loading of biocomposites at room and low temperatures that can be observed in the aviation industry. The fiber optic sensors (FOS) that can outperform electrical sensors in challenging operational environments were used. Standard features were extracted from AE measurements, and a convolutional autoencoder (CAE) was applied to extract deep features from AE signals. Different machine learning methods including discriminant analysis (DA), neural networks (NN), and extreme learning machines (ELM) were used for the construction of classifiers. The analysis is focused on the classification of extracted AE features to classify the source material, to evaluate the predictive importance of extracted features, and to evaluate the ability of used FOS for the evaluation of material behavior under challenging low-temperature environments. The results show the robustness of different CAE configurations for deep feature extraction. The combination of classic and deep features always significantly improves classification accuracy. The best classification accuracy (80.9%) was achieved with a neural network model and generally, more complex nonlinear models (NN, ELM) outperform simple models (DA). In all the considered models, the selected combined features always contain both classic and deep features.

## 1. Introduction

The increasing consumption of lightweight materials, such as fiber-reinforced composites, coincides with expanding regulation to reduce carbon dioxide (CO_2_) emissions and enhance fuel efficiency in transportation [1,2]. The reinforcing fibers can be grouped into synthetic fibers and natural fibers. The synthetic fiber-reinforced polymer (FRP) composites are high-strength materials with excellent fiber-matrix adhesion and could be applied in the automotive, naval, and aerospace industries, etc. [3,4]. However, the main drawbacks of synthetic fibers are high costs, the recycling issues, and the accumulation of waste from the aerospace and automotive sectors [5,6]. In the last few years, natural biocomposite materials have been considered as a substitute for synthetic fiber FRP composites [7,8]. Biocomposites possess satisfactory mechanical properties and are inexpensive, lightweight, and structurally efficient materials [9]. Sometimes porosity can produce a favorable effect in achieving comparatively lightweight materials [10]. Flax (Linum usitatissimum) is one of the most widely used natural fibers. Flax is also one of the first to be extracted, spun, and woven into fabrics. Like cotton, flax fiber is a cellulose polymer, but its structure is more crystalline, making it stronger and more easily wrinkled. Its chemical structure contains cellulose (64–75%), hemicellulose (11–20.6%), pectin (1.8–2.3%), lignin (2–3.0%), and wax (1.5–1.7%) and moisture content is 7.9–10% [11,12]. Flax composites have the potential to be the next-generation materials for structural applications for infrastructure, the automotive industry, and consumer applications in hybrid composite materials.

Environmental concerns have caused a rapid increase in the research of bio-based polymers in recent years. Composite manufacturers are aiming to replace petroleum-based epoxy materials, such as bisphenol A (BPA, 4,4′-(propane-2,2-diyl)diphenol)), with polymers derived from natural sources. Bio-based epoxies are a relatively new class of bio-sourced resins produced by the epoxidation of renewable precursor sources such as unsaturated vegetable oils, saccharides, tannins, cardanols, terpenes, rosins, and lignin [12]. For example, epoxidized soybean oil (ESO) is commercially available and it represents a good replacement for the petroleum-based product. The process of curing involves changing the properties of a given resin by crosslinking the ingredients into a solid form with a hardener. Most common hardeners are petroleum-based, such as isophorone diamines (IPDA), polyamines, polyamides, and anhydrides which represent additional environmental and health issues. However, a lot of attention is given to developing alternative curing agents from biomass. For example, Liu et al. [13] investigated the performance of rosin-based anhydrides as an alternative to petrochemical curing agents. Rosin is abundantly available as the secretion of pine trees or also can be obtained by the distillation of tall oil. The result showed that the rosin-based curing agents’ curing behaviors and mechanical properties were similar to those of the commercial petroleum-based curing agents.

Acoustic emission (AE) monitoring is one of the most suitable methods to detect damage occurrence and its evolution in real-time during the loading of fiber-reinforced polymers [14]. AE monitoring is a passive and non-invasive technique that can be used during the operation of a product or a structure and supplies real-time information that cannot be collected with other techniques. Irreversible processes are producing elastic waves that propagate in the material. The basic task of attached sensors is, therefore, to detect surface displacements (typically out-of-plane) and to generate an electrical or optical signal. High sensitivity of acoustic emission monitoring offers detection of different damage sources, such as transverse matrix cracking as the first occurring damage mechanism in composite laminate [15], delamination, fiber failure, interfacial debonding, debonding between matrix and fibers, fiber pull-out, and friction [16,17,18,19]. Different sources cause a sudden release of energy and produce high-frequency elastic waves. AE waves in a plate-like structure propagate with different modes of Lamb waves that can be detected by different types of AE sensors [20].

Recently, optical fiber sensors (FOS) have rising importance for the detection of acoustic waves in different types of materials. Thus, the research of AE is directed from the field of electro-acoustic sensing technology to photoacoustic sensing technology. Electrical AE systems usually use piezoelectric sensors to detect high-frequency elastic waves in the material. They have some limitations that narrow down their application in harsh environments. FOS in comparison to conventional piezoelectric sensors offers simplified integration with little interference into a wide variety of structures of different materials, resistance to electromagnetic interference, lower weight, and broader sensing capabilities, such as strain, pressure, and temperature in addition to AE [21,22,23]. The components of optical fiber systems are optical sources, i.e., laser, laser diode or LED, optical fibers, that sense elements to transduce the measurement to an optical signal, an optical detector, and a processing unit in the form of an oscilloscope or optical analyzer [24]. One of the classifications of FOS is based on the modulation of the light by a different physical condition. This can classify FOS into scattering-based, intensity-based, polarization-based, phase-based sensors, and wavelength-based sensors [21]. Phase-based sensors use the principle of interferometry and are among the most sensitive FOS. For the signal acquisition, different interferometer configurations were proposed, i.e., Michelson, Mach–Zehnder, Fabry–Perot, Sagnac, Twyman–Green, and Rayleigh. In the case of Michelson interferometer (MI), a beam splitter separates entrance light into two components like in Mach–Zehnder interferometer (MZI). We can find Michelson interferometers for measuring different physical quantities such as AE, stain, temperature, pressure, displacement, and refractive index [25]. MI, similar to MZI, incorporates a directional coupler with sensing and reference fibers. The interferometer sensitivity is dependent on the length of the exposed fiber. Zhang et al. [26] detected AE generated from the partial discharge of high-voltage cable system accessories. The amount of change of the optical fiber length was connected to the AE pressure wave and accompanying pressure change. The phase change is related to axial stretching of the optical fiber and to effective change in fiber diameter which was proven to be negligible in Zhang’s research. Cheng Shi et al. [27] proposed a vibration detection system based on the φ-OTDR (optical time-domain reflectometry) and the optical fiber Michelson interference to detect the acoustic emission in gas-insulated lines. Williams et al. [28] used FOS with Michelson interferometer to convert frequency shift that encodes strain signals into intensity modulation for crack detection in fatigue testing of Al panels.

Different damage mechanisms occur almost simultaneously during composite loading [29,30]. This presents a scientific challenge to assign a specific set of AE signal features to a particular damage mechanism. Often, various machine learning techniques [17,31,32,33] are used in combination with feature selection [34] and Hilbert–Huang transform to extract frequency descriptors [35]. For the automatic extraction of the intrinsic characteristics of signals, deep learning methods are becoming increasingly popular. While feature values are traditionally obtained by manual construction and require certain professional knowledge [36,37], deep learning is used to automatically extract deep features, avoiding the loss of signals in the time and frequency domains during the manual features extraction [38]. Some commonly used deep learning approaches include autoencoders (AEs), deep belief networks (DBNs) [39], convolutional neural networks (CNNs) [40], and recurrent neural networks (RNNs) [41]. A combination of CNN and AE, convolutional autoencoder (CAE) [42], enables the extraction of useful higher-level, yet compact, representation of patterns in an image. CNN consists of translation-invariant convolutional kernels, which serve as feature generators. Such kernels are appropriate when image patterns are expected to be found irrespective of their relative position and such property is desired for recognition tasks when dealing with normal images. The CNN approach has already been successfully applied for deep feature extraction based on AE signals for the characterization of fiber epoxy composites [14] and identification of AE sources for structural health monitoring applications [43].

In this paper, the research was designed to investigate the AE signals at biocomposites having the potential to be the next-generation materials for structural applications for infrastructure, automotive and aviation industries, and consumer applications in hybrid composite materials. There are very few available results in the field of capturing AE signals in the testing of biocomposites, and even less in the field of loading at low temperatures that can be observed in the aviation industry. For the acquisition of AE signals under loading we used OptimAE sensor (developed and commercialized by Optics11) which is one of the world’s first fiber optic acoustic emission sensing systems. The fiber optic sensors (FOS) configured in a Michelson interferometry setup that can outperform electrical sensors in challenging operational environments are used.

This study combines several established methods (acoustic emission measurements, feature extraction, feature selection, machine learning) and adds novelties in the feature extraction stage (deep features) and in the application of these methods in a unique way that provides new perspectives for the characterization of the loaded composites. Different machine learning (ML) methods, such as discriminant analysis (DA), neural networks (NN), and extreme learning machines (ELM), were used in this study for the construction of classifiers of extracted AE features of acquired signals. These methods were combined with the forward feature selection (FFS) procedure and 5-fold cross-validation (CV) to select the most informative extracted AE features and to provide an estimation of the generalization performance. The proposed signal processing structure is focused on the classification of AE signals to recognize the source material, evaluate the predictive importance of extracted classic and deep features of acquired AE signals during loading of biocomposites, and evaluate the ability of used fiber optic AE sensors for evaluation of material behavior under challenging low-temperature environments.

## 2. Materials and Methods

In the research, we used flax biocomposite specimens with a total of eight plies of twill weave flax fabrics. The specimens of 20 × 4 × 200 mm^3^ in size were cut out of the plate of 300 × 300 mm with OMAX 2652A abrasive waterjet cutter. The cutting parameters were as follows: pump jet pressure 3000 bar, diameter of water nozzle 0.3 mm, abrasive Garnet mesh 80, consumption 0.45 kg/min, cutting speed 1023 mm/min. For the preparation of the plate, we used eight plies of flax fabrics 270 × 270 mm^2^, of areal density of 223 g/m^2^ and the average thread diameter was 445 μm. Fabrics were stacked on a polished surface of aluminum alloy 5083 mold with dimensions of 300 × 300 × 10 mm^3^. The surface of a mold was coated with Formula 5 Release Wax Film (by Rexco). Inlet and outlet tubes were placed at each end of the plate. Peel-ply was placed on the top of the dry flax fabrics, followed by placing a vacuum polyethylene infusion mesh (Diatex, OM 70) over the top. Around the fabrics, a vacuum sealing tape, 2.5 × 12 × 15 mm (by R&G Faserverbundwerkstoffe) was placed to seal between bagging film and mold and around inlet and outlet tubes. Vacuum bagging film (PO120, R&G Faserverbundwerkstoffe) was placed over the top of the plate and manually sealed with compression against the vacuum sealing tape. The inlet tube was connected to a vacuum manometer and the outlet tube was connected to the vacuum pump (Value, model V-i240SV, max. vacuum level 99.997%). The mold was placed on a flat aluminum heater to reach the temperature of 40 °C for faster resin flow. The 300 g of GreenPoxy 56 and 108 g of SD 7561 hardener were mixed with a mechanical stirrer at 500 rev/min for 5 min. The GreenPoxy 56 resin is based on the latest innovation in green chemistry. Resin is produced with a high carbon content of plant origin. SR GreenPoxy 56 molecular structure is bio-based at almost 51%. The mechanical properties of epoxy resin are given in Table 1. With its initial viscosity of 700 mPa·s it is made for contact laminating, injection molding, and filament winding. The final glass transition temperature of resin after curing is between 78–85 °C. The vacuum infusion took 5 min to fully impregnate flax fibers. After infusion, the inlet and outlet side were sealed with clams and the mold was inserted into a laboratory autoclave for curing. The air pressure was set to 7 bar and the temperature of 40 °C was established for 8 h. After 8 h, a ramp of 0.5 °C/min was set to reach the temperature of 80 °C for additional 8 h to fully cure the matrix. The cross-sectional view of the flax biocomposite specimen is shown in Figure 1a, where individual longitudinal and transversal flax-bundles with the average diameter of 440 µm are embedded in the epoxy matrix. The fiber volume fraction of vacuum-infused flax composite is 28%. Additionally, to flax biocomposite specimens, we used pultruded glass fiber epoxy GFE, shown in Figure 1b.

Pultrusion is a continuous process to manufacture a unidirectional composite part with constant cross-section, which was, in our case, with the same dimensions as biocomposite samples (20 × 4 × 200 mm^3^). The fiber volume fraction of the GFE composites corresponded to 50% of the overall volume with the average glass fiber diameter of 5 μm.

### 2.1. Experimental Setup

The experimental setup is shown in Figure 2. Biocomposite samples were used in a 3-point bending test on a Messphysic loading machine. The loading cell has a capacity of 50 kN; loading speed was set to 0,06 mm/s. The supports had a span of 80 mm and the loading pin had a diameter of 8 mm. The specimen with a K-type thermocouple attached to the surface was pre-tempered (precooled) before insertion in the chamber in tiny dry ice pellets (solid form of CO_2_) for 1 h to a temperature of the app. −80 °C. The bottom part of the climatization chamber was filled also with a tiny dry ice pellet to help to maintain the adequate temperature of the supports, specimen, and atmosphere in the chamber. The selected testing temperature of the inserted specimens of −80 °C in the climatization chamber with 30-milimeter extruded polystyrene foam walls was controlled by a regulation of the input flow of the liquid nitrogen gas. Before the application of a load, the specimens were maintained at the selected testing temperature for at least 10 min. Five test repetitions were performed for each individual material and environmental condition.

The optical setup consisted of a 4-channel OptimAE system with 2 optical AE sensors (version 05) with a resonance frequency of 215 kHz. The operating principle of the OptimAE system produced by Optics11 is based on interferometry, where two fibers of the same length are configured in a Michelson interferometry setup. Both the sensing fiber and the reference fiber in such a setup are packaged inside the acoustic emission sensor. The reference fiber is coiled around a damper and is isolated from vibration and mechanical disturbances, and the sensing fiber is coiled around a mandrel with a flat bottom. The sensor is attached to the structure with the sensitive mandrel in direct contact with its surface for optimum transmission of the surface acoustic wave to the sensing fiber. The sensor is fixed in position using silicon grease and a rubber clamp. Upon the passing of the acoustic wave through the optical sensor, the vibration is transferred to the sensing mandrel and then to the coiled fiber around it. The resulting interferometric signal is transferred to the OptimAE readout box for signal acquisition, demodulation of the acoustic emission (AE) signal, and communication to a computer with OptimAE software for acoustic event detection and further processing [22]. The threshold for signal acquisition was set to 0.6 nm to avoid noise at low temperatures.

### 2.2. Feature Extraction

Two types of features of acquired AE signals were extracted and compared in analysis: standard and deep features. Standard features are those which are conventionally used to characterize acoustic emission signal data for machine learning tasks. Deep features, on the other hand, were extracted with a deep learning-based method—convolutional autoencoder.

#### 2.2.1. Standard Features

The 13 standard features, which were used for the classification task, are denoted and defined as follows:c1Peak amplitude [nm]—burst signal linear peak amplitude.c2Burst signal rise-time [µs]—elapsed time after the first threshold crossing and until the burst signal maximum amplitude.c3Burst signal duration [µs]—elapsed time after the first and until the last threshold crossing of a burst signal.c4Burst signal energy [au].c5RMS background noise [µV].c6Counts—number of positive threshold crossings [/].c7Spectral centroid [Hz].c8Frequency of the max. amplitude of Fourier transform spectrum [Hz].c9Frequency of the max. amplitude of continuous wavelet transformation (using the complex Morlet wavelet) [Hz].c10Partial power of Fourier spectrum between 0 and 75 kHz [/].c11Partial power of Fourier spectrum between 75 and 150 kHz [/].c12Partial power of Fourier spectrum between 150 and 300 kHz [/].c13Partial power of Fourier spectrum between 300 and 475 kHz [/].

Surface acoustic waves in the material result in optical path length changes in the interferometer arms and because of that optical signal peak amplitudes c1 can be represented with units of length, compared to electrical signals induced by the piezoelectric effect in the PZT sensors.

#### 2.2.2. Convolutional Autoencoder and Deep Features

Autoencoders learn, in an unsupervised manner, a way to reconstruct (decode) compressed (decoded) data so that the output’s representation matches its original input most efficiently. This encourages the identification of the most important patterns in data, which, extracted as deep features, may also aid classification tasks. Typical autoencoders operate on data in the forms of 1D feature vectors, which is why, for extracting deep features from 2D data (images), convolutional autoencoder (CAE) is needed.

For the acquisition of deep features, two convolutional autoencoders were used (denoted as CAE-1 and CAE-2). CAE-1 and CAE-2 differ in kernel shapes and strides used in convolutional and max-pooling (C+P) layers. These hyperparameters stayed the same for all of the specific CAE configurations. A detailed explanation of the CAE structure (especially CAE-1) is provided in [14]. CAE structures are summarized in Table 2.

Each input data (scalogram image) are initially (in the CAE) split into 4 smaller images, resulting in 4 partial images of the same length and ¼ times the height. There are two consecutive C+P (1st and 2nd layers), which operate on each of these 4 bands separately. Then, before the final C+P layer (latent layers), all resulting feature maps are concatenated back. Two consecutives fully connected (FC) and a final bottleneck FC round up the model’s encoder. Then, in the model’s decoder, the same number of layers are used (including the two FC layers), just in the reversed order. After the latent section, the feature maps are split back into stripes, which are subjugated separately to two consecutive up-sampling and transposed convolution layers, before finally being merged back to a single image. After training the CAE, deep features are extracted from the model’s encoder and denoted as: d1, d2, etc. Deep features are extracted automatically and, therefore, do not have physical meaning but are designed to minimize information loss of the input–output mapping of the CAE.

Training data consisted of AE scalograms, obtained via wavelet transformation (complex Morlet wavelets type). Pre-processing operations used on the 2D arrays (scalograms) in Model 2 were, in order: absolute value, matrix-wise standardization, squaring, and division by max. value. Performing these steps ensured stable model training, reduced overall noise, and preserved distinct feature patterns in CWT scalograms. The frequency range of scalograms was 0–100 kHz (up to a tenth of sampling frequency). For the training of CAE, the loss function chosen was squared error, whereas ADAM [44] was used as the optimization algorithm.

For classification analysis purposes, there were several different model configurations with varying hyperparameters. These configurations are denoted as “s1-s2-s3-s4-s5-s6-s7”, which stand for:s1Number of kernels used in 1st convolutional layers.s2Number of kernels used in 2nd convolutional and 1st transposed convolutional layers.s3Number of kernels used in convolutional and transposed convolutional layers of the latent section.s4Number of neurons in 2nd and 4th FC layers.s5Number of neurons in the bottleneck (3rd FC) layer. This number also represents the number of extracted deep features per input image.s6Number of training epochs.s7The batch size of training samples.

The complete dataset for this study is composed of 12 files (in Excel format), each representing either CAE-1 or CAE-2 autoencoders and different configurations (s1-s2-s3-s4-s5-s6-s7). Each file contains extracted standard and deep features for each AE measurement and the class descriptor denoting biocomposite and glass fiber epoxy composite. See Appendix A associated with this study.

### 2.3. Machine Learning Methods

Several established machine learning (ML) methods were used in this study for the construction of classifiers of extracted AE features with respect to the origin of the substrate (biocomposite or GFE). The classification methods comprise discriminant analysis (DA), neural networks (NN), and extreme learning machines (ELM). These methods were combined with the forward feature selection (FFS) procedure and 5-fold cross-validation (CV) to:(1)Select the most informative extracted AE features;(2)provide estimation of generalization performance (based on CV).

The forward feature selection procedure provides a simple framework for the selection of the most relevant subset of features which maximizes the evaluation criterion (classification accuracy in our study). The FFS is an iterative method that starts with the evaluation of each feature and selects a feature that results in the best evaluation criterion. Next, all possible combinations of the selected feature and a subsequent feature are evaluated, and a second feature is selected, and so on until the maximum of the evaluation criterion is reached.

The evaluation criterion is calculated by using a cross-validation procedure which prevents overfitting and provides a context for estimating the generalization performance of the study (see also Section 2.4 for a description of the 5-fold CV approach).

The classification methods are introduced in the following sections.

#### 2.3.1. Discriminant Analysis

Discriminant analysis (DA) as a set of methods offers distinguishment of acquired AE signals based on extracted features into categories with calculated discrimination boundaries of population and determines how to allocate new observations [45]. The assumption with the DA is that different categories of data can be based on different Gaussian distributions. The results of previous research confirmed that DA can be considered as one of the top classifiers, which is probably due to a bias–variance trade-off, with simple linear or quadratic boundaries and stable estimates obtained by Gaussian models [46]. In this study, linear DA (LDA) and quadratic DA (QDA) were applied as benchmark methods providing insight into the complexity of the decision problem (comparatively with respect to other nonlinear ML methods).

#### 2.3.2. Neural Networks

Neural networks (NNs) are among the most frequently used models for machine learning tasks [47]. Like in many ML models, the single-valued predictor *y* (output) can be represented by a generic expression of the inputs *x*, which may be *K*-dimensional $x=\left\{x_{1},x_{2},\ldots ,x_{K}\right\}$. There are *N*_h_ neurons in the hidden layer. The model can be described as
(1)$$y=f_{o}\left(\overset{N_{h}}{\underset{j=0}{\sum }}w_{j}f_{h}\left(\sum_{i=0}^{K}w_{ji}x_{i}\right)\right)$$
where *f*_h_ is the hidden layer activation function, *f*_o_ represents the output layer activation function, *w_ji_* are hidden layer weights, and *w_j_* are the output layer weights. In this study, simple NN architecture with 4 hidden neurons in a single hidden layer was applied. Sigmoid activation functions were used in the hidden layer, whereas a linear activation function in the output layer. All NN models were trained by Levenberg–Marquardt training algorithm, combined with Bayesian regularization (LM–BR) to prevent overfitting [48].

#### 2.3.3. Extreme Learning Machines

Extreme learning machines (ELMs) are learning algorithms designed for single hidden layer feedforward networks [49]. It can provide good generalization performance at very fast learning speeds. In essence, the algorithm is based on random hidden layer nodes selection and analytical determination of the output weights. Various applications of ELM are summarized in [50]. The ELM model can be defined as follows:(2)$$y=\overset{N_{h}}{\underset{j=1}{\sum }}w_{j}f\left(x\right)$$
where $w_{j}$ represents the output weights, and *f* the nonlinear feature mapping nonlinear piecewise continuous function that satisfies the ELM universal approximation theorems [51,52]. Sigmoid, hyperbolic tangent, Gaussian, and multi-quadratic functions are among the commonly used ELM mapping functions. The ELM model training consists of two main stages. In the first stage (random feature mapping), once hidden layer weights are randomly initialized, input data are transformed into an ELM feature space using mapping function (f). In the second stage, the weights $w_{j}$ connecting the hidden layer and the output layer are solved by minimizing the approximation error (mean squared error). In this study, a hidden layer was constructed using *N_h_* = 100 neurons with sigmoid activation functions. Additional settings are not required for this method.

### 2.4. Objectives and Evaluation Criterion

The objectives of this research are focused on the classification of AE signals to recognize the source material either as biocomposite or GFE composite with a use of fiber optical AE sensors. The evaluation criterion (also referred to as “performance criterion”) in this study is defined as classification accuracy, which denotes the percentage of correctly classified samples (*N_c_*) with respect to the number of all available samples (*N*), as follows:(3)$$Accuracy=100\frac{Nc}{N}\;\left[\%\right]\;$$

The particular objectives of this study include:The evaluation of the predictive importance of extracted classic and deep features.The comparative analysis of various ML-based classifiers to construct the discriminative predictors.

The objectives are investigated by the following steps which define our research framework:Extraction of classic features (c1, c2,…, c10) from the acquired AE signals.Analysis of various CAE configurations and the extraction of deep features (d1,d2,…).Initiation of a 5-fold cross-validation loop for the analysis of classifiers:i.Split the data, consisting of extracted AE features (classic features + deep features), into 5 subsets.ii.Train each classifier on 4 subsets and test the generalization performance on the remaining 1 subset.iii.Repeat the procedure through all 5 subsets and average the generalization performance (classification accuracy) over all 5 subsets.iv.Repeat the CV-based analysis for all applied classifiers: LDA, QDA, NN, and ELM.Summarize the results concerning the chosen ML-based predictors and concerning selected AE features.

## 3. Results

Load-deflection curves for the flax biocomposite and GFE sample are shown in Figure 3. At room temperature, the GFE samples reached an average maximum flexural load *P_max_* of 2721 N or 1020 MPa of flexural strength. The flax composite samples reached the average flexural load *P_max_* of 388 N or 145 MPa of flexural strength. GFE samples have more than 7-times higher flexural strength, 4.8-times higher flexural modulus than that of the biocomposite, and similar deflections or breaking strain of 2.7% in flax composite and 2.5% in GFE composite. The GFE yields to linear load-deflection response, whereas the flax leads to a curved nonlinear load-deflection curve due to the lower strength of flax fibers and high-volume resin content (72%).

At the temperature of −80 °C (which is considered also as a high cryogenic temperature), the flexural stress and the breaking strains decrease as the resin becomes harder and more brittle. Contrary, the flexural modulus significantly increases at −80 °C. The reason is the reduction in polymer chain mobility, which increases the binding forces between the molecules. This, in turn, leads to a higher number of AE signals, detected in the elastic region of loading for both tested material groups, which can be depicted in Figure 3. Lower temperature for GFE samples led to a 6.6% *P_max_* decrease and a more significant 47% decrease in flexural deflection at fracture. In the case of biocomposites, the influence of temperature reduction led to a similar 7.4% *P_max_* decrease and a 38% decrease in deflection, i.e., smaller than with GFE. With the onset of viscoelastic behavior in flax biocomposites, the stiffness has reduced with the loading progression and specimen degradation with micro-cracks in the resin. This stiffness reduction is higher with an increased load with biocomposites in comparison to GFE composites. With the increase in load, we can expect fiber-matrix debonding and micro-crack length increase that can lead to delamination. AE activity rises toward the point of catastrophic failure and attains high values after the reach of maximum flexural load with pronounced signals of fiber pull-out and fiber breakage. The lower fiber volume fraction of flax composite together with higher fiber diameter compared to GFE composites results in lower AE activity around *P_max_*. At room temperatures, we can notice the highest AE amplitude values at the moment of catastrophic failure, while in the case of lower temperatures, the amplitudes values remain high also after the reach of maximum flexural load. The fracture of GFE after the linear force-deflection response is sudden and fast, accompanied by a high number of strong AE signals as the fibers are breaking, whereas the fracture of flax biocomposites occurs over the wider time-frame with a lower number of AE signals acquired. The average flexural properties, strength, modulus, and breaking strains for flax and GFE composites at room temperature and at −80 °C are given in Table 3.

The classification results of acquired AE signals are summarized in Table 4. The first two columns denote different CAE configurations, as explained in Section 2.2.2. The next four columns provide results expressed as CV-based classification accuracy, obtained by applying different ML-based classifiers: LDA, QDA, NN, and ELM. The last column (Mean) provides the average accuracy, calculated across all applied classifiers.

Based on the results in Table 4, the following conclusions can be summarized regarding the CAE configurations and ML models:The results, in terms of CV-based accuracy, are comparable across different CAE configurations, thus confirming the robustness of CAE-based deep feature extraction. The best result (NN, 80.9%) is achieved with the CAE-2 configuration “32-64-20-512-8-100-256”, but all results are very similar and do not differ significantly. The best average result corresponds to the configuration CAE-1 “16-32-14-256-6-100-128”.A comparison of machine learning models shows that with more complex nonlinear models (NN and ELM models), higher classification accuracy can be achieved (compared to simple LDA and QDA models). This confirms that the application of nonlinear ML models can be useful in the AE-based characterization of composite materials. The best results are achieved with the NN model, which is a neural network with one hidden layer of four neurons, and the ELM models with one hundred hidden neurons show only slightly lower performance.

The more specific results, revealing the selection of AE-based features, are shown in Figure 4 and Figure 5. The results are taken from the CAE-1 configuration “16-32-14-256-6-100-128” which provides the overall best average result. Results for different CAE configurations are similar and are, therefore, not shown.

Combined features, sequentially selected by the FFS method, are denoted as labels on the *x*-axis. Plots show accuracy for different feature sets: only classic features, only deep features, and combined features. Labels of selected features for each feature set are shown in corresponding colors (blue for combined features, red for classic features, and magenta for deep features). Figure 4 shows results for the LDA model (representing a simple linear classifier), and Figure 5 shows results obtained by an NN model which represents a nonlinear method with overall best classification results. Results for QDA are similar to LDA, and ELM results are similar to NN results (See Table 4) and are, therefore, not shown. In both examples (LDA and NN), the improvement of the combined feature set over classical (or deep features only) is evident. In both examples, various classic features, such as c9 (FVTA), c1 (peak amplitude), etc., are selected together with deep features (d1, d2, etc.).

These results provide evidences regarding the feature selection as follows:A comparison of separate use of only classic or only deep features shows that only classic features allow better classification compared to only deep features.The combination of classic and deep features always significantly improves classification accuracy.The selected combined features (using the FFS method) in all the considered models always contain both classic and deep features.Often, one of the deep features is the first chosen feature (the most informative feature).The order of the selected deep features is different for each CAE configuration because the feature extraction process is “unsupervised” and can result in different solutions.

## 4. Conclusions

The research was focused on the evaluation of acquired AE signals during the loading of flax biocomposite specimens and more frequently used pultruded GFE specimens at room and low temperatures that can be observed in the aviation industry. Since the temperatures of −80 °C present a limitation for conventional electro-acoustic sensing technology with piezoelectric sensors, the phase-based FOS configured in a Michelson interferometry setup was used.

For the comparative analysis of AE signals, standard and deep features were extracted with a comparison of two different convolutional autoencoders (CAE) as deep learning-based methods. For the construction of classifiers of extracted AE features, different ML-based methods (LDA, QDA, NN, and ELM) were used. The results with respect to CAE configurations, feature extraction, and ML-based classification can be summarized as follows:The classification accuracy of ML-based classifiers is comparable across different CAE configurations, thus confirming the robustness of CAE-based deep feature extraction.The combination of both types of features—classic and deep features—always significantly improves classification accuracy, and the selected combined features in all the considered models always contain both classic and deep features.Nonlinear models (NN and ELM models) provide higher classification accuracy (compared to the simple LDA model); therefore, the application of nonlinear ML models is useful for the classification of the source material (biocomposite or GFE composite).The best classification result (80.9% accuracy of classification of source material) is achieved with an NN classifier, combined features, and CAE-2 configuration “32-64-20-512-8-100-256”.

The results of the proposed approach show that phase-based FOS has the potential for practical use in structural health monitoring of FRP composites exposed to harsh environmental conditions (e.g., aviation applications). The proposed signal processing structure with selected deep autoencoder architecture and machine learning method offers improved classification accuracy with the use of combined extracted features.

This study was conducted with a set of laboratory experiments and examined samples of biocomposites and glass fiber epoxy composites. The results are indicative of the potential of the proposed methodology for the characterization of loaded composite materials, yet further research is needed with a wider range of examined composite structures and different loading and environmental conditions. Further research is also envisioned in the area of the application of fiber optic sensors in comparison with electrical sensors in challenging operational environments to find out the best operating conditions for each type of AE sensor.

## Figures and Tables

**Figure 1 sensors-22-06886-f001:**
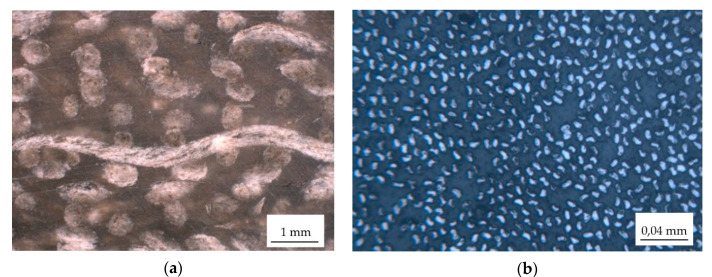
Cross-sectional view of the (**a**) flax biocomposite and (**b**) GFE specimens.

**Figure 2 sensors-22-06886-f002:**
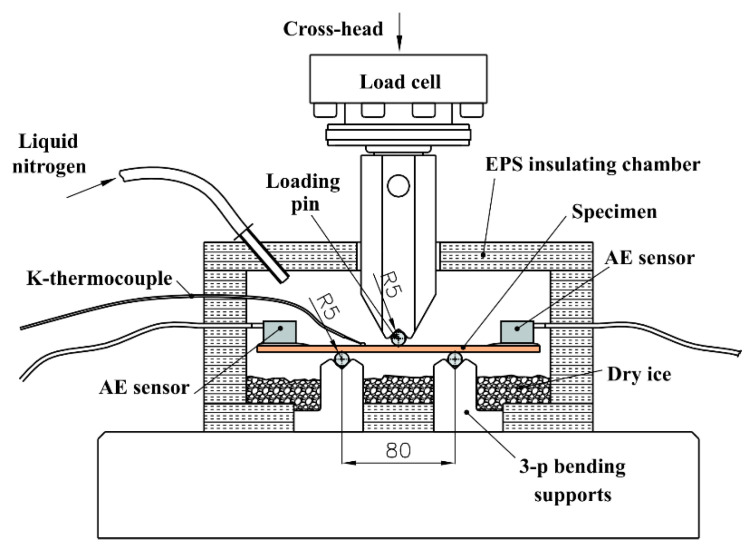
Experimental setup.

**Figure 3 sensors-22-06886-f003:**
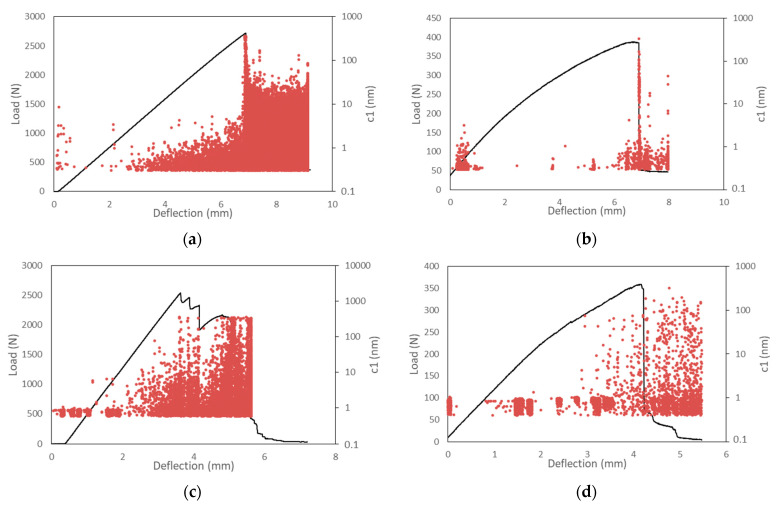
Load-deflection curve with peak amplitude values for GFE and flax biocomposite specimens: (**a**) GFE, room temperature, (**b**) flax biocomposite, room temperature, (**c**) GFE at the temperature of −80 °C, and (**d**) flax biocomposite at the temperature of −80 °C.

**Figure 4 sensors-22-06886-f004:**
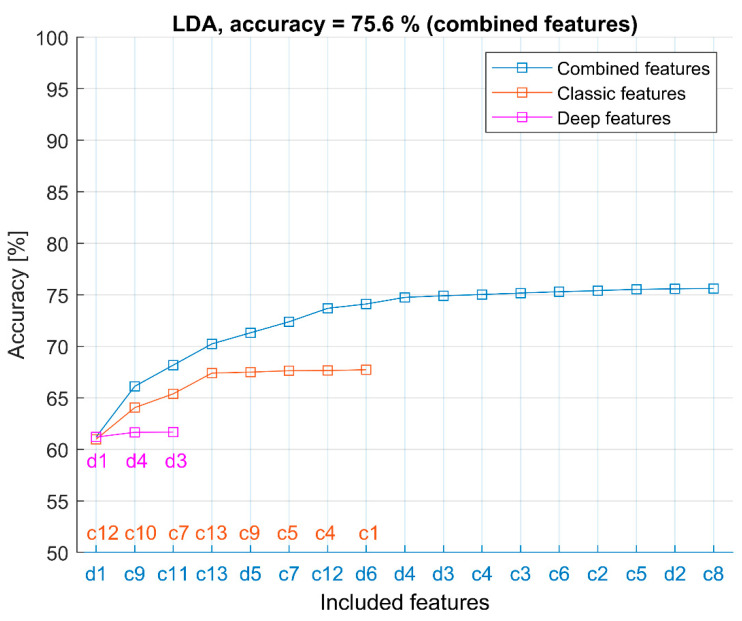
LDA classifier and selected combined features (d1, c9, c11, c13, d5, …) in CAE-1 configuration “16-32-14-256-6-100-128”.

**Figure 5 sensors-22-06886-f005:**
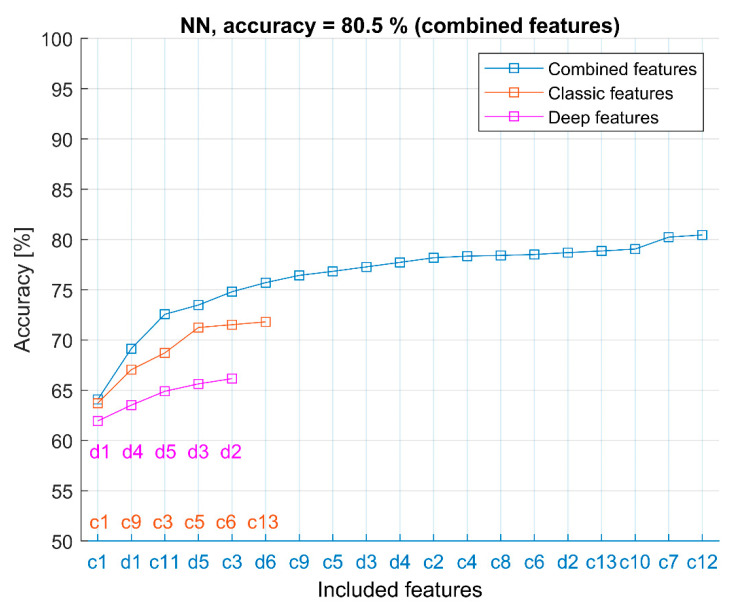
NN classifier and selected combined features (c1, d1, c11, d5, c3, …) in CAE-1 configuration “16-32-14-256-6-100-128”.

**Table 1 sensors-22-06886-t001:** Properties of plant-based epoxy SR GreenPoxy 56/SD 7561 after 8 h of post-curing at 80 °C.

Property/Curing Cycle	24 h, 20 °C + 8 h, 80 °C
Tensile strength (MPa)	68
Young modulus (MPa)	2980
Elongation (%)	6.4
Charpy Impact toughness (kJ/m^2^)	52
Glass transition temperature (°C)	78

**Table 2 sensors-22-06886-t002:** Comparison of CAE models regarding kernel shapes and strides.

CAE Model	CAE-1		CAE-2	
Input scalogram shape (height × length)	48 × 432		32 × 512	
Layer’s fixed hyperparameters *	Kernel shape	Stride	Kernel shape	Stride
(1st) Convolution Layers	3 × 5	1 × 3	3 × 3	1 × 1
(1st) Max-Pooling Layers	2 × 2	2 × 2	2 × 2	2 × 2
(2nd) Convolution Layers	3 × 5	1 × 3	3 × 3	1 × 1
(2nd) Max-Pooling Layers	2 × 2	2 × 2	2 × 4	2 × 4
(Latent) Convolution Layer	3 × 3	1 × 1	2 × 64	1 × 1
(Latent) Max-Pooling Layer	2 × 2	2 × 2	2 × 2	2 × 2
5 Fully-connected Layers	/	/	/	/
(Latent) Up-Sampling Layer	2 × 2	2 × 2	2 × 2	2 × 2
(Latent) Transposed Convolution Layer	3 × 3	1 × 1	2 × 64	1 × 1
(1st) Up-Sampling Layers	2 × 2	2 × 2	2 × 4	2 × 4

* Fixed hyperparameters (i.e., kernel shapes and strides) in classification analysis for either CAE were not changed in any of the CAE configurations (as opposed to hyperparameters “s1-s2-s3-s4-s5-s6-s7”).

**Table 3 sensors-22-06886-t003:** Flexural properties of flax and GFE composites at room temperature and at −80 °C.

Mechanical Properties	Designation	Temperature (°C)	Flax Composite	GFE Composite
Flexural strength	*σ*_f_ (MPa)		145	1020
Flexural modulus	*E*_f_ (GPa)	20 °C	8.6	41.6
Flexural strain at break	*ε*_B_ (%)		2.7	2.5
Flexural strength	*σ*_f_ (MPa)		135	952
Flexural modulus	*E*_f_ (GPa)	−80 °C	11.3	79
Flexural strain at break	*ε*_B_ (%)		1.5	1.2

**Table 4 sensors-22-06886-t004:** Classification results provide classification accuracy, summarized over various CAE configurations, and classified by LDA, QDA, NN, and ELM classifiers.

CAE Model	Deep Autoencoder	Classification Accuracy [%]				
	Architecture	LDA	QDA	NN	ELM	Mean
CAE-1	12-24-10-256-6-100-256	75.5	75.3	80.1	79.8	77.7
	16-32-14-256-6-100-128	75.6	76.1	80.5	79.9	78.0
	30-60-30-512-12-65-32	75.2	74.6	80.4	79.2	77.4
	32-64-20-512-8-110-20	75.3	75.0	80.5	79.4	77.6
	32-64-26-512-10-110-64	75.2	74.7	80.8	79.3	77.5
	32-74-30-432-14-65-32	71.7	72.3	78.8	79.5	75.6
	34-68-34-648-16-65-32	75.0	75.2	77.7	79.3	76.8
	38-76-34-564-16-45-32	75.2	74.4	78.2	79.6	76.9
CAE-2	16-32-14-256-6-100-256	74.5	75.5	80.2	79.8	77.5
	32-64-16-256-6-75-256	73.7	75.6	80.4	78.2	77.0
	32-64-20-512-8-100-256	75.8	75.8	80.9	78.7	77.8
	6-12-12-128-4-75-356	74.0	73.3	77.4	77.9	75.7

## Data Availability

The data presented in this study are openly available in Mendeley Data at http://dx.doi.org/10.17632/zs82gfkkr9.1.

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
