# Peer review of "Characterization of Biocomposites and Glass Fiber Epoxy Composites Based on Acoustic Emission Signals, Deep Feature Extraction, and Machine Learning"

_sensors, 2022, doi:10.3390/s22186886_

Round 1

Reviewer 1 Report

In this paper, the authors are focused on the evaluation of acquired AE (acoustic emission) signals during loading of flax biocomposite specimens and pultruded GFE (glass fiber epoxy) specimens at room and low temperatures.

From my point of view there are some aspects to improve:

1. Abstract does not contain any quantitative data about the results. The abstract should be revised and improved.

2. New methods and protocols should be described in detail while well-established methods can be briefly described and appropriately cited.

3.  The manufacturing method of biocomposite samples was described in detail, but the characterization of the materials is missing. A microscopic analysis of the flax biocomposite and pultruded GFE specimens should be taken.

An example of microstructures analysis of glass-fiber-reinforced polymer (GFRP) was presented in the paper mentioned below.

You can cite the paper: Sabău, Emilia, Razvan Udroiu, Paul Bere, Ivan Buranský, and Cristina-Ştefana Miron-Borzan. 2020. "A Novel Polymer Concrete Composite with GFRP Waste: Applications, Morphology, and Porosity Characterization" Applied Sciences 10, no. 6: 2060. https://doi.org/10.3390/app10062060

4. How many samples (flax biocomposite and pultruded GFE) have been analyzed within experiments? Please specify it within the manuscript. Usually a minimum 5 samples should be analyzed in order to evaluate the repeatability of the experiments.

5. The specimens were water-jet cut. Please specify what type of machine was used and some parameters of the cutting process.

6. Are the limitations of this study noted? The limitations of this study should be discussed.

Author Response

Dear Reviewer,

Thank you for providing us with new suggestions and directions with the paper review. We invested a lot of our time and effort in improvements to the paper and addressing your suggestions. Please see the attachment for the response.

Reviewer 2 Report

This is review for ‘Characterization of biocomposites and glass fiber epoxy composites based on AE signals, deep feature extraction and machine learning’ by Tomaž Kek et al. This work was highlighting the results of AE measurements in the loading of biocomposites at room and low temperatures. Moreover, this paper showed the proposed signal processing structure is focused on the classification of AE signals to recognize the source material. Overall, the work appears precisely performed and interpreted.  To improve this work, however, reviewer would like to recommend minor revision.  Before accepting for publication, the following comments need to be addressed.

-      The abstract is quite long, however, no critical points about this study was found. The abstract needs to include discussion about major objective; the summary of the results; and major findings. What is the relevance and novelty of this study?

-      In the introduction part, references are partly missing and the fundamental knowledge about acoustic emission measurements is completely missing. Reviewer would like to recommend providing technical details and fundamental concepts of acoustic emission measurements.

-      Reference format is not consistent: some have issue number, but some others have not. Some have not page information. Some have volume number, but some others have not. Some have DOI information, but some others have not.

Author Response

Dear Reviewer,

Thank you for providing us with new suggestions and directions with the paper review. We invested a lot of our time and effort in improvements to the paper and addressing the your suggestions. Please see the attachment for the response.

Reviewer 3 Report

Very interesting and rigorous work, but must be considerably improved.

1. Line 300. You mention the use of the forward feature selection  (FFS) procedure but do not explain it at all. You must introduce it and explain it extensively.

2. Line 374 and on. Starting the Results section you introduce the Deflection Curves without apparent motivation. Why do you introduce them now? What do they mean in relation to the aim of the paper? What do mean those curves that suddendly drop down? At what levels of the physical magnitudes (horizontal and vertical axes) and why? A brief introduction of the motivation of using them now is mostly needed.

3. Line 414. SFS method? Or FFS method? You must explain that method very clearly to let the reader understand the procedures you are following.

4. Lines 437-443. It seems you are following the "FFS method" and even let many details as supposedly understood by the reader, but he/she cannot follow you unless you explain what the FFS method means at this point; all in the case you are using that specific method which is not clear at all. Please do not let anything to be a-priori undesrtood by the reader.

5. Lines 470 and on. You take for granted that the results of the classification experiments mean that they have a relationship and are useful to structural health monitoring but for the reader there is not such relationship at all. Why classifiyng materials into GFE and bio-composite materials should give us information about dammage mechanisms in parts exposed to environmental conditions?

And most importantly: to what extent? Should you conduct some more related experiments in order to obtain a realistic criterion to such monitoring of materials? You really need to attend a number of practical issues regarding this matrerial monitoring in practice after the results or your classification experiment or experiments if additional classifications are needed.

If there is such monitoring possibility you shoud indeed explain it in a much better and much more extended way. 

Once you have achieved this, additional practical explanations about the use of specific classification(s) in practical materials monitoring is strongly needed.

6. The accuracy results of the classification are not high at all. However, you should provide the results in a more neutral and informative classification performance measure, such as weighted F-measure for instance.

Author Response

Dear Reviewer,

Thank you for providing us with new suggestions and directions with the paper reviews. We invested a lot of our time and effort in improvements to the paper and addressing the reviewer’s suggestions. This is also the first time we received as many as five review reports. In this paper, we wanted to present a study that combines several established methods (acoustic emission measurements, feature extraction, feature selection, machine learning) and adds novelties in the feature extraction stage (deep features) and in the application of these methods in a unique way that provides new perspectives for the characterization of the loaded composites. The proposed signal processing structure is focused on the classification of AE signals to recognize the source material, evaluate the predictive importance of extracted classic and deep features of acquired AE signals during loading of biocomposites, and evaluate the ability of used fiber optic AE sensor for evaluation of material behavior under challenging low-temperature environment.

We also prepared the dataset to be published upon acceptance of the paper. Reserved DOI: 10.17632/zs82gfkkr9.1. The dataset was referenced in the paper (appendix A) and can be previewed at:

https://data.mendeley.com/datasets/zs82gfkkr9/draft?a=a1e235c0-0c84-4c85-bcd7-51ccb6429da1

We hope that we have adequately supplemented the paper following the recommendations and that the article will be accepted for publication.

Reviewer 4 Report

In the Title, please use Acoustic emission instead of AE as the abbreviation.

Result section, from Line 374 to 382, the font size is different.

In sections 2.3.2 and 2.3.3, what is the y signifies? Accuracy!

Figure 2 is not well explained in the text. What is C1 stand for? Add the room and lower temperature in the figure, and mention the lower temperature.

In Line 313, the authors mentioned they used QDA. Where do the authors use QDA in this text? Where is the Accuracy % curve?

The manuscript text does not elucidate much about Figure 3 and Figure 4. It is not clear from the text. Why the d1, c9, and c11 have deep features accuracy %, and others do not? Similarly, d1 to d6 have classical features accuracy %, but all others have combined elements. Please explain it clearly in the text according to the authors' figures. How is the accuracy % calculated? Also, the accuracy% calculation formula is not well defined as how the accuracy is calculated in the case of QDA and LDA analysis.

In line 400 and Line 411, the CAE1 configuration fonts are different. Please follow the original fonts.

Author Response

Dear Reviewer,

Thank you for providing us with new suggestions and directions for the paper review. We invested a lot of our time and effort in improvements to the paper and addressing your suggestions. Please see the attachment for the response.

Reviewer 5 Report

The paper focuses on the evaluation of acquired AE signals in the process of loading flax bio-composite specimens at a room temperature and at an extremely low temperature for the aviation industry. Bio-composites with flax fibbers could be next generation materials in automotive and aviation industries. Using of these lightweight materials could reduce carbon dioxide (CO2) emissions. So, the work solve up-to-date problem, is interesting and with good results. Results summarized in the conclusion are valuable.

I have some comments to the manuscript: 

-          The machine learning methods were used for the comparative analysis of AE signals. But the reasons for the selection of methods LDA, QDA, NN and ELMs are not presented in the paper.  The more detailed explanation of the functioning of methods LDA and QDA should be added. Are they belonging to methods of neural networks learning and deep learning? From the Table 3, it can be derived, that yes.

-          The keyword “characterization” is too general to be a keyword.

-          Authors mentioned a combination of machine learning methods with other methods, and between them was also an unsupervised learning (page 3, lines 112-113). But unsupervised learning is a part of machine learning.

-          The list of standard features is presented in the paper (lines 229-244). But I have not found similar list of deep features in the paper. According to Figure 3 and Figure 4, deep features d1, …, d6 were used in experiments.

-          It is not clear from the description of Figure 2 what each of the 4 parts of the picture exactly illustrates. I suggest explicit labeling (for example a) b) c) and d)) and a precise description of each part.

-          The Table 3 header is on page 9 and the rest of the table is on the next page 10.

-          Can the authors of the article provide the dataset on which the models were trained? Or at least can they provide an illustration of part of the data with a table?

-          It is not clear what the individual results in Figure 3 and Figure 4 represent. What can we read from x-axis? That some models were trained on a group of attributes and some on single attributes? Why such an ordering of the features on x-axis?

-          Some shortcuts should be defined (GFE, FOS, …). It would be helpful for readers of the paper.

-          The Conclusion section only recapitulates what was done and the achieved results. There is no mention of future research.

Author Response

(The authors gave the same response as above.)

Round 2

Reviewer 1 Report

All the comments are addressed well and utilized to improve the manuscript. The manuscript is acceptable.